# Scaling Off-Policy Reinforcement Learning with Batch and Weight Normalization

**Daniel Palenicek**[1,2] **Florian Vogt**[3] **Joe Watson**[4] **Jan Peters**[1,2,5,6]
[1]Technical University of Darmstadt [2]hessian.AI [3]University of Freiburg [4]University of Oxford
[5]German Research Center for AI (DFKI) [6]Robotics Institute Germany (RIG)
daniel.palenicek@tu-darmstadt.de

## Abstract

Reinforcement learning has achieved significant milestones, but sample efficiency remains a bottleneck for real-world applications. Recently, CrossQ has demonstrated state-of-the-art sample efficiency with a low update-to-data (UTD) ratio of 1. In this work, we explore CrossQ's scaling behavior with higher UTD ratios. We identify challenges in the training dynamics, which are emphasized by higher UTD ratios. To address these, we integrate weight normalization into the CrossQ framework, a solution that stabilizes training, has been shown to prevent potential loss of plasticity and keeps the effective learning rate constant. Our proposed approach reliably scales with increasing UTD ratios, achieving competitive performance across 25 challenging continuous control tasks on the DeepMind Control Suite and MyoSuite benchmarks, notably the complex `dog` and `humanoid` environments. This work eliminates the need for drastic interventions, such as network resets, and offers a simple yet robust pathway for improving sample efficiency and scalability in model-free reinforcement learning.

## 1 Introduction

Reinforcement Learning (RL) has shown great successes in recent years, achieving breakthroughs in diverse areas. Despite these advancements, a fundamental challenge that remains in RL is enhancing the sample efficiency of algorithms. Indeed, in real-world applications, such as robotics, collecting large amounts of data can be time-consuming, costly, and sometimes impractical due to physical constraints or safety concerns. Thus, addressing this is crucial to make RL methods more accessible and scalable.

Different approaches have been explored to address the problem of low sample efficiency in RL. Model-based RL, on the one hand, attempts to increase sample efficiency by learning dynamic models that reduce the need for collecting real data, a process often expensive and time-consuming [39, 22, 12, 17]. Model-free RL approaches, on the other hand, have explored increasing the number of gradient updates on the available data, referred to as the update-to-data (UTD) ratio [33, 10], modifying network architectures [4], or both [9, 18, 20, 32]. A central tension in these research directions is balancing the sample efficiency of the agent against the computational complexity, i.e., wall-clock time, of the underlying algorithm. Algorithmic adjustments such as model-based rollouts, computing auxiliary exploration rewards and higher UTDs can all significantly increase the wall-clock time of the algorithm. Likewise, architectural changes such as larger models [32] and more ensemble members [9] also increase the wallclock time of the method. Ideally, we desire architectural and algorithmic changes that balance sample efficiency and *simplicity*, such as CrossQ [4], which showed that careful use of batch normalization unlocks significantly greater sampler efficiency of deep actor-critic methods without significantly impacting the wallclock time.

39th Conference on Neural Information Processing Systems (NeurIPS 2025).

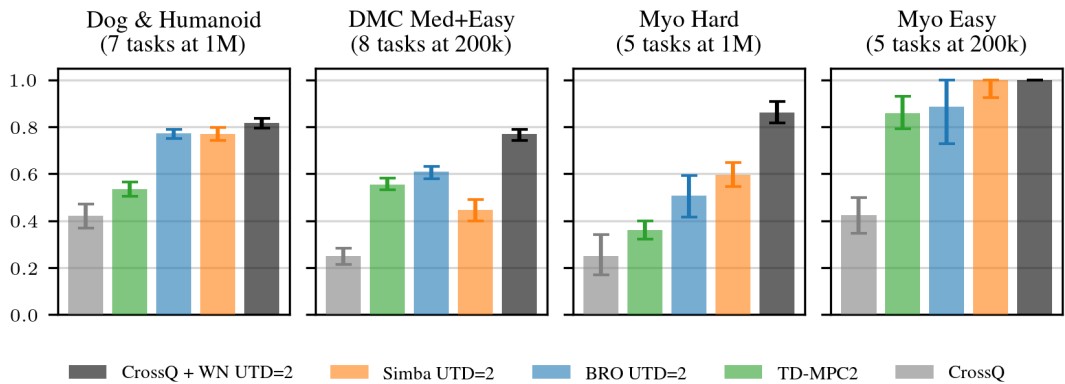

Figure 1: *CrossQ +* WN UTD=2 *outperforms* SIMBA UTD=2 *and* BRO UTD=2. In comparison, our proposed CrossQ + WN is a simple algorithm that, unlike BRO, does not require extra exploration policies or full parameter resets. We present results for 25 complex continuous control tasks from the DMC and MyoSuite benchmarking suites. 1.0 marks the maximum score achievable on the respective benchmarks (DMC *return* up to 1000 / MyoSuite up to 100% *success rate*). We present IQM and 90% stratified bootstrap confidence intervals aggregated over multiple environments and 10 seeds each.

In this work, we build upon CrossQ [4], the model-free RL algorithm that showed state-of-the-art sample efficiency on the MuJoCo [42] continuous control benchmarking tasks, and also enabled learning omni-directional locomotion policies in 8 minutes of real-world experience [6]. Notably, the authors achieved this by carefully utilizing batch normalization (BN, Ioffe [21]) within the actor-critic architecture. A technique previously thought not to work in RL, as reported by Hiraoka et al. [18] and others. The insight that Bhatt et al. [4] offered is that one needs to carefully consider the different state-action distributions within the Bellman equation and handle them correctly to succeed. This novelty allowed CrossQ at a low UTD of 1 to outperform the then state-of-the-art algorithms that scaled their UTD ratios up to 20. Even though higher UTD ratios are more computationally expensive, they allow for larger policy improvements using the same amount of data. This naturally raises the question: *How can we extend the sample efficiency benefits of CrossQ and* BN *to the high* UTD *training regime?* Which we address in this manuscript.

**Contributions.** In this work, we show that the vanilla CrossQ algorithm is brittle to tune on DeepMind Control (DMC) and MyoSuite environments and can fail to scale reliably with increased compute. To address these limitations, we propose the addition of weight normalization (WN),

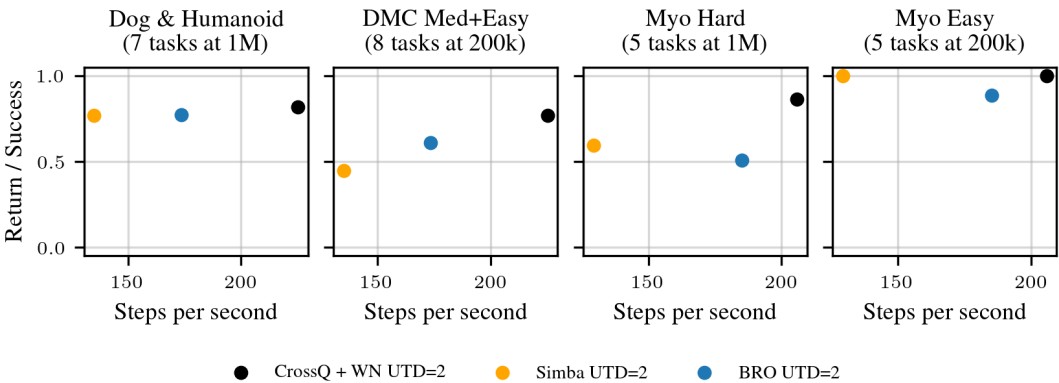

Figure 2: Comparing performance against wall clock time, measured in environment steps per second (so larger is better) on a single RTX 4090 workstation, we observe the CrossQ + WN outperforms SIMBA and BRO across all environments. We present results for 25 complex continuous control tasks from the DMC and MyoSuite benchmarking suites. 1.0 marks the maximum score achievable on the respective benchmarks (DMC *return* up to 1000 / MyoSuite up to 100% *success rate*).

which we show to be a simple yet effective enhancement that stabilizes CrossQ. We motivate the combined use of WN and BN on insights from the continual learning and loss of plasticity literature and connections to the effective learning rate. Our experiments show that incorporating WN not only improves the stability of CrossQ but also allows us to scale its UTD, thereby significantly enhancing sample efficiency.

## 2 Preliminaries

This section briefly outlines the required background knowledge for this paper.

**Reinforcement learning.** A Markov decision process (MDP) [35] is a tuple $\mathcal{M} = \langle \mathcal{S}, \mathcal{A}, \mathcal{P}, r, \mu_0, \gamma \rangle$, with state space $\mathcal{S} \subseteq \mathbb{R}^n$, action space $\mathcal{A} \subseteq \mathbb{R}^m$, transition probability $\mathcal{P} : \mathcal{S} \times \mathcal{A} \to \Delta(\mathcal{S})$, the reward function $r : \mathcal{S} \times \mathcal{A} \to \mathbb{R}$, initial state distribution $\mu_0$ and discount factor $\gamma$. We define the RL problem according to Sutton and Barto [40]. A policy $\pi : \mathcal{S} \to \Delta(\mathcal{A})$ is a behavior plan, which maps a state $s$ to a distribution over actions $a$. The discounted cumulative return is defined as $\mathcal{R}^\pi(s, a) = \sum_{t=0}^{\infty} \gamma^t r(s_t, a_t)$, where $s_0 = s$, $a_0 = a$ and $s_{t+1} \sim \mathcal{P}(\cdot | s_t, a_t)$ and $a_t \sim \pi(\cdot | s_t)$ otherwise. The Q-function of a policy $\pi$ is the expected discounted return $Q^\pi(s, a) = \mathbb{E}[\mathcal{R}^\pi(s, a)]$. The goal of an RL agent is to find an optimal policy $\pi^*$ that maximizes the expected return from the initial state distribution $\pi^* = \arg\max_\pi \mathbb{E}_{s \sim \mu_0} [Q^\pi(s, a)]$.

Actor-critic methods address this optimization problem by jointly learning neural network representations for the Q-function and the policy. The policy network is optimized to maximize the Q-values, while the Q-function is optimized to minimize the squared Bellman error, where the value target is computed by taking an expectation over the learned Q function

$$V(s_{t+1}) = \mathbb{E}_{a_{t+1} \sim \pi_\theta(\cdot | s_{t+1})} [Q_{\bar{\theta}}(s_{t+1}, a_{t+1})]. \tag{1}$$

To stabilize the Q-function learning, Haarnoja et al. [14] found it necessary to use a target Q-network in the computation of the value function instead of the regular Q-network. The target Q-network is structurally equal to the regular Q-network, and its parameters $\bar{\theta}$ are obtained via Polyak Averaging over the learned parameters $\theta$. While this scheme ensures stability during training by explicitly delaying value function updates, it also arguably slows down online learning [34, 23, 31].

Instead of relying on target networks, CrossQ [4] addresses training stability issues by introducing batch normalization (BN, Ioffe [21]) in its Q-function and achieves substantial improvements in sample and computational efficiency over Soft actor-critic (SAC). A central challenge when using BN in Q networks is distribution mismatch: during training, the Q-function is optimized with samples $s_t, a_t$ from the replay buffer. However, when the Q-function is evaluated to compute the target values (Equation (1)), it receives actions sampled from the current policy $a_{t+1} \sim \pi_\theta(\cdot | s_{t+1})$. Those samples have no guarantee of lying within the training distribution of the Q-function. BN is known to struggle with out-of-distribution samples, as such, training can become unstable if the distribution mismatch is not correctly accounted for [4]. To deal with this issue, CrossQ removes the separate target Q-function and evaluates both Q values during the critic update in a single forward pass, which causes the BN layers to compute shared statistics over the samples from the replay buffer and the current policy. This scheme effectively tackles distribution mismatch problems, ensuring that all inputs and intermediate activations are effectively forced to lie within the training distribution.

**Normalization techniques in RL.** Normalization techniques are widely recognized for improving the training of neural networks, as they generally accelerate training and improve generalization [19]. There are many ways of introducing different types of normalizations into the RL framework. Most commonly, authors have used layer normalization (LN) within the network architectures to stabilize training [18, 30, 27]. Recently, CrossQ has been the first algorithm to successfully use BN layers in RL [4]. The addition of BN leads to substantial gains in sample efficiency. In contrast to LN, however, one needs to carefully consider the different state-action distributions within the critic loss when integrating BN. In a different line of work, Hussing et al. [20] proposed the integration of unit ball normalization and projected the output features of the penultimate layer onto the unit ball in order to reduce Q-function overestimation.

**Increasing update-to-data ratios.** Although scaling up the UTD ratio is an intuitive approach to increase the sample efficiency, in practice, it comes with several challenges. Nikishin et al. [33]

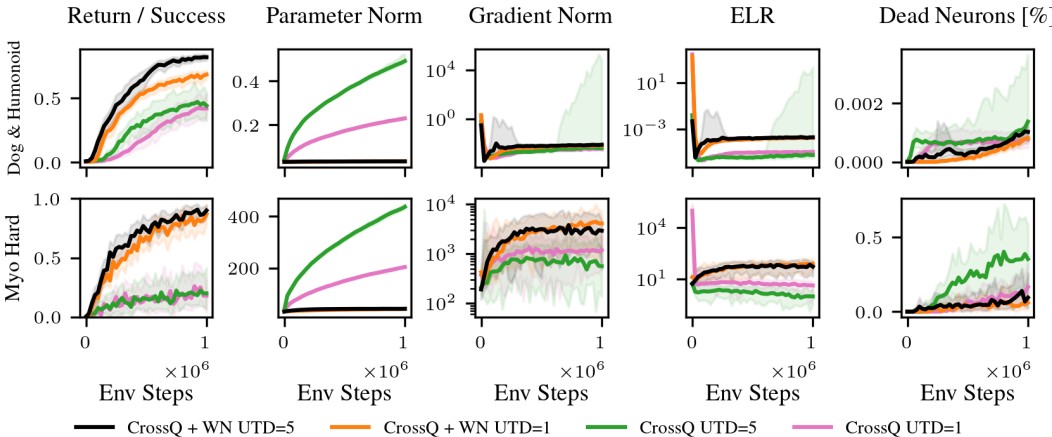

Figure 3: *Growing parameter norms hinder learning.* The performance benefits of CrossQ fail to scale to more complex, higher dimensional tasks such as humanoid locomotion and muscular manipulation. Investigating this, we find that the critic parameter norms increase significantly with increasing UTD ratios. As a result, the effective learning rate (ELR) drops and the number of dead neurons increases. Regularizing the critic parameters with weight norm (WN), we successfully mitigate the parameter norm growth and therefore maintain a more consistent ELR, leading to better performance on these more complex tasks. Uncertainty quantification depicts the 90% stratified bootstrap confidence intervals.

demonstrated that overfitting on early training data can inhibit the agent from learning anything later in the training. The authors dub this phenomenon the primacy bias. To address the primacy bias, they suggest to periodically reset the network parameters while retraining the replay buffer. Many works that followed have adapted this intervention [10, 32]. While often effective, regularly resetting is a very drastic intervention and by design induces regular drops in performance. Since the agent has to start learning from scratch repeatedly, it is also not very computing efficient. Finally, the exact reasons why parameter resets work well in practice are not well understood [28]. Instead of resetting there have also been other types of regularization that allowed practitioners to train stably with high UTD ratios. Janner et al. [22] learn a world model and use it to generate synthetic data which is used in a high UTD setting to stabilize training. REDQ [9] leverages sampling from an ensemble of Q-functions, while Hiraoka et al. [18] use dropout and LN to effectively scale to higher UTD ratios.

## 3 Batch normalization alone fails to scale with respect to task complexity

Bhatt et al. [4] demonstrated CrossQ's state-of-the-art sample efficiency on the MuJoCo task suite [42], while at the same time also being very computationally efficient. However, on the more extensive DMC and MyoSuite task suites, we find that CrossQ requires tuning. We further find that it works on some, but not all, environments stably and reliably.

Figure 3 shows CrossQ training performance on the DMC dog and humanoid and the Myo Hard tasks aggregated by environment suite. We plot the IQM and 90% confidence intervals for each metric across 10 seeds. The figure compares a CrossQ with UTDs 1 and 5, where the hyperparameters were identified through a grid search over learning rates and network sizes, as detailed in Table 1. The first shows the agent's training performance, while the other columns show network parameter norms, gradient norms, the effective learning rate [44], and the fraction of dead neurons. Here, we identify three different training behaviors. We notice that CrossQ does not benefit from higher UTD ratios, but performance remains similar on the provided tasks. Overall, for all CrossQ runs we notice large confidence intervals.

**Growing network parameter norms.** The second column of Figure 3 displays the sum over the L2 norms of the dense layers in the critic network. This includes three dense layers, each hidden layer with a dimension of 512. On both task suites, CrossQ exhibits growing network weights over the course of training. We find that the effect is particularly pronounced for CrossQ

with increasing UTD ratios. Growing network weights have been linked to a loss of plasticity, a phenomenon where networks become increasingly resistant to parameter update, which can lead to premature convergence [11]. Additionally, the growing magnitudes pose a challenge for optimization, connected to the issue of growing activations, which has recently been analyzed by Hussing et al. [20]. Further, growing network weights decrease the effective learning rate when the networks contain normalization layers [43, 30].

In summary, the scaling results for vanilla CrossQ are mixed. While increasing UTD ratios is known to yield increased sample efficiency, if careful regularization is used [22, 9, 33], CrossQ alone with BN cannot benefit from it. We notice that with increasing UTD ratios, CrossQ's weight layer norms grow significantly faster and overall larger. This observation motivates us to further study the weight norms in CrossQ to increase UTD ratios.

## 4 Combining batch normalization and weight normalization enables scaling

Inspired by the combined insights of Van Hasselt et al. [43] and Lyle et al. [30], we propose to integrate CrossQ with weight normalization (Salimans and Kingma [36], WN) as a means of counteracting the rapid growth of weight norms we observe with increasing update-to-data (UTD) ratios. A weight normalized parameter $\tilde{w}$ is constrained to have an L2 norm of $c$, an additional hyperparameter,

$$\tilde{w} = c\,w\,/\,||w||_2. \tag{2}$$

Our approach is based on the following reasoning: Due to the use of BN in CrossQ, the critic network exhibits scale invariance, as previously noted by Van Laarhoven [44].

**Theorem 1 (Van Laarhoven [44])** *Let $f(X; w, b, \gamma, \beta)$ be a function, with inputs $X$ and parameters $w$ and $b$ and $\gamma$ and $\beta$ batch normalization parameters. When $f$ is normalized with batch normalization, $f$ becomes scale-invariant with respect to its parameters, i.e.,*

$$f(X; cw, cb, \gamma, \beta) = f(X; w, b, \gamma, \beta), \tag{3}$$

*with scaling factor $c > 0$.*

The proof is provided in Appendix A.

This property allows us to introduce WN as a mechanism to regulate the growth of weight norms in CrossQ without affecting the critics outputs. Further, it can be shown, that for such a scale invariant function, the gradient scales inversely proportionally to the scaling factor $c > 0$.

**Theorem 2 (Van Laarhoven [44])** *Let $f(X; w, b, \gamma, \beta)$ be a scale-invariant function. Its gradient*

$$\nabla f(X; cw, cb, \gamma, \beta) = \nabla f(X; w, b, \gamma, \beta)/c, \tag{4}$$

*scales inversely proportional to the scaling factor $c \in \mathbb{R}$ of its parameters $w$.*

The proof is provided in Appendix B.

Recently, Lyle et al. [30] demonstrated that the combination of LN and WN can help mitigate loss of plasticity. Since the gradient scale is inversely proportional to $c$, keeping norms constant helps to maintain a stable effective learning rate (ELR, Van Hasselt et al. [43]), further enhancing training stability. We conjecture that maintaining a stable ELR could also be beneficial when increasing the UTD ratios in continuous control RL. As the UTD ratio increases, the networks are updated more frequently with each environment interaction. Empirically, we find that the network norms tend to grow quicker with increased UTD ratios (Figure 3), which in turn decreases the ELR even quicker and could be the reason for lower training performance. From this observation, we hypothesize that the training phenomena that affect plasticity also appear when attempting sample-efficient learning with higher UTDs. This hypothesis suggests that regularization techniques for plasticity could also be used to achieve more sample-efficient RL. As a result, we empirically investigate the effectiveness of combining CrossQ with WN with increasing UTD ratios.

**Implementation details.** We apply WN to all hidden linear layers of the network, ensuring that their weights remain unit norm after each gradient step by projecting them onto the unit ball, similar to Lyle et al. [30]. The reason for not placing WN on the output layer is to allow scaling of predicted

Q values to arbitrary magnitudes and crucially avoid the risk of underfitting. Additionally, layers that are not followed by BN are not scale invariant w.r.t. re-scaling their weights. While we could employ a learning rate schedule [30] we did not investigate this here as this would add additional complexity. Additionally, we impose weight decay on all parameters that remain unbounded—specifically, the final dense output layer. In practice, we use AdamW [29] with a decay of 0 (which falls back to vanilla Adam [24]) for the normalized intermediate dense layers and $1e{-}2$ otherwise.

**Target networks.** CrossQ removes the target networks from the actor-critic framework and showed that using BN training remains stable even without them [4]. While we find this to be true in many cases, we find that especially in DMC, the re-integration of target networks can help stabilize training overall (see Section 5.4). However, not surprisingly, we find that the integration of target networks with BN requires careful consideration of the different state-action distributions between the $s, a$ and $s', a' \sim \pi(s')$ exactly as proposed by Bhatt et al. [4]. To satisfy this, we keep the joined forward pass through both the critic network as well as the target critic network. We evaluate both networks in *training mode*, i.e., they calculate the joined state-action batch statistics on the current batches. As is common, we use Polyak-averaging with a $\tau = 0.005$ from the critic network to the target network.

# 5 Experimental results

To evaluate the effectiveness of our proposed CrossQ + WN method, we conduct a comprehensive set of experiments on the DeepMind Control Suite [41] and MyoSuite [7] benchmarks. For DMC we report individual results for the hard (`dog` and `humanoid`), as well as Medium+Easy (`cheetah-run`, `walker-run`, `hopper-stand`, `finger-turn-hard`, `quadruped-run`, `fish-swim`, `hopper-hop`, `pendulum-swingup`) due to their varying difficulties. Equally, we split MyoSuite hard and easy environments. Our primary goal is to investigate the scalability of CrossQ + WN with increasing UTD ratios and to assess the stabilizing effects of combining CrossQ with WN. We compare our approach to several baselines, including the recent BRO [32], CrossQ [4], TD-MPC2 [16], and SIMBA [27], a concurrent approach utilizing layer norm. Figure 6 in the appendix further provides a SR-SAC [10] baseline, a version of SAC [14] with high UTD ratios and network resets.

## 5.1 Experimental setup

Our implementation is based on the SAC implementation of `jaxrl` codebase [25]. We implement CrossQ following the author's original codebase and add the architectural modifications introduced by [4], incorporating batch normalization in the actor and critic networks. We extend this approach by introducing WN to regulate the growth of weight norms and prevent loss of plasticity and add target networks. We perform a grid search to focus on learning rate selection and layer width.

We evaluate 25 diverse continuous control tasks, 15 from DMC and 10 from MyoSuite. These tasks vary significantly in complexity, requiring different levels of fine motor control and policy adaptation with high-dimensional state spaces up to $\mathbb{R}^{223}$. Each experiment is run for 1 million environment steps and across 10 random seeds to ensure statistical robustness. We evaluate agents every $25,000$ environment steps for 5 trajectories. For the DMC Medium&Easy and Myo Easy we plot the first $200k$ steps, as all methods learn much faster than the 1 million steps. As proposed by Agarwal et al. [1], we report the interquartile mean (IQM) and $90\%$ stratified bootstrap confidence intervals (CIs) of the return (or success rate, respectively), if not otherwise stated. For the BRO and SIMBA baseline results, for computational reasons, we take the official evaluation data that the authors provide. The official BRO codebase is also based on `jaxrl`, and the authors followed the same evaluation protocol. All experiments were run on a compute cluster with `RTX 3090` and `A5000` GPUs, where all 10 seeds run in parallel on a single GPU via `jax.vmap`.

## 5.2 Weight normalization allows CrossQ to scale to harder tasks effectively

We provide empirical evidence for our hypothesis that controlling the weight norm and, thereby, the ELR can stabilize training (Figure 3). We show that through the addition of WN, CrossQ + WN shows stable training and can stably scale with increasing UTD ratios.

Figure 7 shows per environment results of our experiments encompassing all 25 tasks evaluated across 10 seeds each. Based on that, Figure 1 shows aggregated performance over all environments

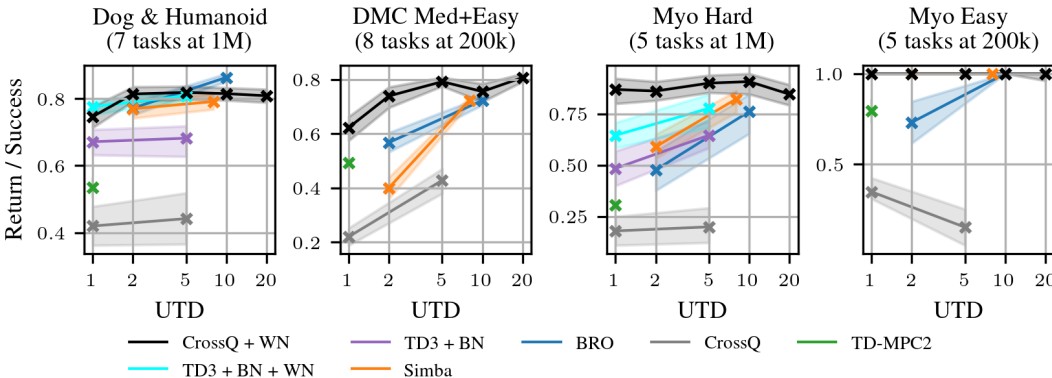

Figure 4: *CrossQ* WN UTD *scaling behavior.* We plot the IQM return and $90\%$ confidence intervals for different UTD ratios. Results are aggregated over 15 DMC environments and 10 random seeds each according to Agarwal et al. [1]. The sample efficiency scales reliably with increasing UTD ratios and remains stable even when there are no more performance gains, which is a crucial property.

from Figure 7 per task suite, with a separate aggregation for the most complex `dog` and `humanoid` environments.

Scaling results in Figure 4 show that CrossQ + WN is competitive to the BRO and SIMBA baselines on both DMC and MyoSuite, especially on the more complex `dog` and `humanoid` tasks already on lower UTD ratios. Notably, CrossQ + WN UTD=5 uses only half the UTD of BRO and does not require any parameter resets and no additional exploration policy. Further, it uses $\sim 90\%$ fewer network parameters—BRO reports $\sim 5M$, while our proposed CrossQ + WN uses only $\sim 600k$ (these numbers vary slightly per environment, depending on the state and action dimensionalities).

In contrast, vanilla CrossQ UTD=1 exhibits much slower learning on most tasks and, in some environments, fails to learn performant policies. Moreover, the instability of vanilla CrossQ at UTD=5 is particularly notable, as it does not reliably converge across environments (Figure 7).

These findings highlight the necessity of incorporating additional normalization techniques to sustain effective training at higher UTD ratios. This leads us to conclude that CrossQ benefits from the addition of WN, which results in stable training and scales well with higher UTD ratios. The resulting algorithm can match or outperform state-of-the-art baselines on the continuous control DMC and MyoSuite benchmarks while being much simpler algorithmically.

## 5.3 Stable scaling of CrossQ + WN with UTD ratios

To visualize the stable scaling behavior of CrossQ + WN we ablate across UTD ratios. Figure 4 shows training performances aggregated over multiple environments for 10 seeds each at 1M steps and 200k steps respectively. We confirm that CrossQ + WN shows reliable scaling behavior. With increasing compute, the performance increases or stays constant which is desirable. We see, that for the same UTD ratio, CrossQ + WN nearly always beats both the BRO and SIMBA baselines. Further, we demonstrate, that the sample-efficiency and stable UTD scaling benefits of BN+WN extend beyond CrossQ and SAC to other base algorithms, such as TD3.

## 5.4 Hyperparameter ablation studies

We also ablate the different hyperparameters of CrossQ + WN, by changing each one at a time. Figure 5 shows aggregated results of the final performances of each ablation. We will briefly discuss each ablation individually.

**Removing weight normalization.** Not performing weight normalization results in the biggest drop in performance across all our ablations. This loss is most drastic on the MyoSuite tasks and often results in no meaningful learning. Showing that, as hypothesized, the inclusion of WN into the CrossQ framework yields great improvements in terms of sample efficiency and training stability,

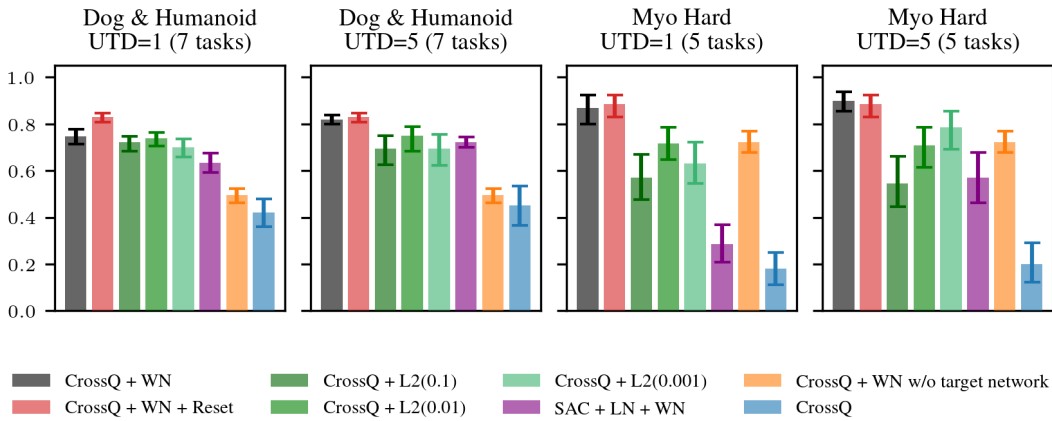

Figure 5: An ablation study comparing CrossQ + WN against a soft L2 penalty on the weights, as well as other design decisions such as target networks. All ablations use a critic network with two hidden layers of dimension 512. The results show that the hard constraint outperforms the soft approach across a range of regularization scales and tasks. Uncertainty quantification depicts the 90% stratified bootstrap confidence intervals.

especially for larger UTD ratios. In general, lower UTD ratios are already reasonably competitive in overall performance.

**Target networks.** Ablating the target networks shows that on MyoSuite, there is a significant difference between using a target network and not using a target network. Results on DMC show a large drop in performance. There, removing target networks leads to a significant drop in performance, nearly as large as removing weight normalization. This finding is interesting, as it suggests that CrossQ + WN without target networks is not inherently unstable. But there are situations where the inclusion of target networks is required. Further investigating the role and necessity of target networks in RL is an interesting direction for future research.

**L2 regularization.** Figure 5 investigates the performance of a soft L2 penalty on the weights compared to the weight normalization proposed in this paper. Across all tasks, and sweeping across soft regularization scalings, weight normalization outperforms the soft L2 penalty. Our hypothesis is that, in principle, soft L2 regularization could work in a similar way to the proposed WN; however, it would require per-task tuning and potentially even scheduling to result in a stable ELR. In comparison, the *hard constraint* via WN guarantees a stable weight norm by design and as such is easier to employ.

**Parameter resets.** Our experiments show that CrossQ + WN is able to scale without requiring drastic interventions such as parameter resets. However, it is still interesting to investigate whether CrossQ + WN's performance could benefit from parameter resets. CrossQ + WN + Reset in Figure 5 investigates this question. We see, that there is a slight improvement on the 7 dog and humanoid tasks, on all other 18 tasks, performance remains the same. The main takeaway is, that CrossQ + WN scales stably *without requiring* parameter resets.

## 6 Related work

RL has demonstrated remarkable success across various domains, yet sample efficiency remains a significant challenge, especially in real-world applications where data collection is expensive or impractical. Various approaches have been explored to address this issue, including model-based RL, UTD ratio scaling, and architectural modifications.

**Update-to-data ratio scaling.** Model-free RL methods, including those utilizing higher UTD ratios, focus on increasing the number of gradient updates per collected sample to maximize learning from available data. High UTD training introduces several challenges, such as overfitting to early training data, a phenomenon known as primacy bias [33]. This can be counteracted by periodically resetting

the network parameters [33, 10]. However, network resets introduce abrupt performance drops. Nauman et al. [32] demonstrated that full parameter resets of the critic can effectively preserve learning capacity using a UTD ratio up to 10. However, such resets inevitably impact the wall-clock time due to relearning function approximation several times during learning. Alternative approaches use an ensemble of Q-functions to reduce overestimation bias that occurs under high UTD ratio training regimes [9]. Due to the decreased computational efficiency of using a large number of Q-functions, Hiraoka et al. [18] propose to replace the ensemble of critics with dropout and layer normalization. Both methods utilize a UTD ratio of 20, which is highly inefficient.

**Normalization techniques in RL.** Normalization techniques have long been recognized for their impact on neural network training. LN [3] and other architectural modifications have been used to stabilize learning in RL [18, 32]. Yet BN has only recently been successfully applied in this context [4], challenging previous findings, where BN in critics caused training to fail [18]. WN has been shown to keep ELRs stable and prevent loss of plasticity [30], when combined with LN, making it a promising candidate for integration into existing RL frameworks. Bjorck et al. [5] show that using spectral normalization (SN) enables training with large scale neural networks in deep RL. SN divides a layers weight matrix by its largest singular value, regularizing the Lipschitz continuity of the function approximation and therefore stabilizing the gradients. BN has been used by Bhatt et al. [4], where it achieves impressive results, but requires slight adjustments when the UTD ratio is scaled. Concurrent work [27] injected 'simplicity bias' into their actor and critic architecture, encouraging the model to favor 'simpler' features for its predictions. Simba incorporates a residual feedforward block and layer normalization in both the actor and critic networks and achieved state of the art results. WN has been shown to keep ELRs stable and prevent loss of plasticity [30], when combined with LN, making it a promising candidate for integration into existing RL frameworks. Ceron et al. [8] recently analyzed the impact of different normalization on the performance of RL algorithms. Hafner et al. [15] designed a vision-based MBRL algorithm Dreamer that leverages world models to master a wide range of diverse tasks, and also relies on several normalization techniques. They utilize root mean square layer normalization (RMSNORM) [46] before the activation function and normalize the returns. To successfully scale deep RL on the Atari 100k benchmark and achieve human-level sample efficiency, Schwarzer et al. [37] rely on regularization techniques. Rather than use normalization methods, they use the shrink-and-perturb method [2] in shallow layers and full parameter resets in deeper layers to preserve network plasticity. To scale to higher UTDs, they introduce AdamW [29] and gradually decrease the number of steps for the computation of the TD error for faster convergence. Lee et al. [26] argue that the loss of plasticity observed in deep RL when the UTD ratio is increased can be mitigated by using LN, sharpness-aware minimization (SAM, Foret et al. [13]), incorporating parameter resets in the last layers and replacing the ReLU activation function with a 'concetenated ReLU' function [38]. Voelcker et al. [45] demonstrated that parameter resets are not strictly necessary when the UTD ratio is increased to improve sample efficiency. They identify the generalization ability of the critic as the main source of training instabilities under high UTD regimes. They demonstrate empirically that architectural regularization can mitigate overestimation and divergence, but it does not guarantee proper generalization. On the other hand, leveraging synthetic data generated by a learned world model can help mitigate the effects of distribution shift, thereby enhancing generalization.

# 7   Conclusion, limitations & future work

In this work, we have addressed the instability and scalability limitations of CrossQ in RL by integrating weight normalization. Our empirical results demonstrate that WN effectively stabilizes training and allows CrossQ to scale reliably with higher UTD ratios. The proposed CrossQ + WN approach achieves competitive or superior performance compared to state-of-the-art baselines across a diverse set of 25 complex continuous control tasks from the DMC and MyoSuite benchmarks. These tasks include complex and high-dimensional `humanoid` and `dog` environments. This extension preserves simplicity while enhancing robustness and scalability by eliminating the need for drastic interventions such as network resets. In this work, we only consider continuous state-action benchmarking tasks. While our proposed CrossQ + WN performs competitively on these tasks, its performance on discrete state-action spaces or vision-based tasks remains unexplored. We plan to investigate this in future work. Moreover, the theoretical basis of our work does not directly connect to the convergence rates or sample efficiency of the underlying RL algorithm, but rather to mitigate observed empirical phenomena regarding the function approximation alone.

## Acknowledgments and Disclosure of Funding

We would also like to thank Tim Schneider, Cristiana de Farias, João Carvalho and Theo Gruner for proofreading and constructive criticism on the manuscript. This research was funded by the research cluster Third Wave of AI, funded by the excellence program of the Hessian Ministry of Higher Education, Science, Research and the Arts, hessian.AI and has benefited from the early stages of the funding by the Deutsche Forschungsgemeinschaft (DFG, German Research Foundation) under Germanys Excellence Strategy EXC-3057; funding will begin in 2026. This work was also supported by a UKRI/EPSRC Programme Grant [EP/V000748/1].

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

# A Proof Scale Invariance

Proof of Theorem 1.

$$f(\boldsymbol{X}; c\boldsymbol{w}, cb, \gamma, \beta) = \frac{g(c\boldsymbol{X}\boldsymbol{w} + cb) - \mu(g(c\boldsymbol{X}\boldsymbol{w} + cb))}{\sigma(g(c\boldsymbol{X}\boldsymbol{w} + cb))}\gamma + \beta \tag{5}$$

$$= \frac{cg(\boldsymbol{X}\boldsymbol{w} + b) - c\mu(g(\boldsymbol{X}\boldsymbol{w} + b))}{|c|\sigma(g(\boldsymbol{X}\boldsymbol{w} + b))}\gamma + \beta \tag{6}$$

$$= \frac{g(\boldsymbol{X}\boldsymbol{w} + b) - \mu(g(\boldsymbol{X}\boldsymbol{w} + b))}{\sigma(g(\boldsymbol{X}\boldsymbol{w} + b))}\gamma + \beta = f(\boldsymbol{X}; \boldsymbol{w}, b, \gamma, \beta) \tag{7}$$

# B Proof Inverse Proportional Gradients

To show that the gradients scale inversely proportional to the parameter norm, we can first write

$$f(\boldsymbol{X}; c\boldsymbol{w}, cb, \gamma, \beta) = \frac{g(c\boldsymbol{X}\boldsymbol{w} + cb) - \mu(g(c\boldsymbol{X}\boldsymbol{w} + cb))}{\sigma(g(c\boldsymbol{X}\boldsymbol{w} + cb))}\gamma + \beta \tag{8}$$

$$= \frac{g(c\boldsymbol{X}\boldsymbol{w} + cb)}{\sigma(g(c\boldsymbol{X}\boldsymbol{w} + cb))}\gamma - \frac{\mu(g(c\boldsymbol{X}\boldsymbol{w} + cb))}{\sigma(g(c\boldsymbol{X}\boldsymbol{w} + cb))}\gamma + \beta. \tag{9}$$

$$\tag{10}$$

As the gradient of the weights is not backpropagated through the mean and standard deviation, we have

$$\nabla_w f(\boldsymbol{X}; c\boldsymbol{w}, cb, \gamma, \beta) = \frac{g'(c\boldsymbol{X}\boldsymbol{w} + cb)X}{|c|\sigma(g(\boldsymbol{X}\boldsymbol{w} + b))}\gamma. \tag{11}$$

$$\tag{12}$$

The gradient of the bias can be computed analogously

$$\nabla_b f(\boldsymbol{X}; c\boldsymbol{w}, cb, \gamma, \beta) = \frac{g'(c\boldsymbol{X}\boldsymbol{w} + cb)}{|c|\sigma(g(\boldsymbol{X}\boldsymbol{w} + b))}\gamma. \tag{13}$$

$$\tag{14}$$

# C Hyperparameters

Table 1 gives an overview of the hyperparameters that were used for each algorithm that was considered in this work.

Table 1: Hyperparameters

| Hyperparameter | CrossQ | CrossQ + WN | Simba | SR-SAC | BRO |
|---|---|---|---|---|---|
| Critic learning rate | 0.0003 | 0.0003 | 0.0001 | 0.0003 | 0.0003 |
| Critic hidden dim | 512 | 512 | 512 | 256 | 512 |
| Actor learning rate | 0.0003 | 0.0003 | 0.0001 | 0.0003 | 0.0003 |
| Actor hidden dim | 256 | 256 | 128 | 256 | 256 |
| Initial temperature | 1.0 | 1.0 | 0.01 | 1.0 | 1.0 |
| Temperature learning rate | 0.0001 | 0.0001 | 0.0001 | 0.0003 | 0.0003 |
| Target entropy | $\lvert A \rvert/2$ | $\lvert A \rvert/2$ | $\lvert A \rvert/2$ | $\lvert A \rvert$ | $\lvert A \rvert$ |
| Target network momentum | 0.005 | 0.005 | 0.005 | 0.005 | 0.005 |
| UTD | 1,2,5,10,20 | 1,5 | 2,8 | 32 | 2,10 |
| Number of critics | 2 | 2 | 1 | 2 | 1 |
| Action repeat | 2 | 2 | 2 | 2 | 1 |
| Discount | 0.99 (DMC) 0.95 (Myo) | 0.99 (DMC) 0.95 (Myo) | 0.99 (DMC) 0.95 (Myo) | 0.99 (DMC) 0.95 (Myo) | 0.99 (DMC) 0.99 (Myo) |
| Optimizer | Adam | AdamW | AdamW | Adam | AdamW |
| Optimizer momentum ($\beta_1$, $\beta_2$) | (0.9, 0.999) | (0.9, 0.999) | (0.9, 0.999) | (0.9, 0.999) | (0.9, 0.999) |
| Policy delay | 3 | 3 | 1 | 1 | 1 |
| Warmup transitions | 5000 | 5000 | 5000 | 10000 | 10000 |
| AdamW weight decay critic | 0.0 | 0.01 | 0.01 | 0.0 | 0.0001 |
| AdamW weight decay actor | 0.0 | 0.01 | 0.01 | 0.0 | 0.0001 |
| AdamW weight decay temperature | 0.0 | 0.0 | 0.0 | 0.0 | 0.0 |
| Batch Normalization momentum | 0.99 | 0.99 | N/A | N/A | N/A |
| Reset Interval of networks | N/A | N/A | N/A | every 80k steps | *at 15k, 50k, 250k, 500k and 750k steps* |
| Batch Size | 256 | 256 | 256 | 256 | 128 |

# D  Aggregated learning curves

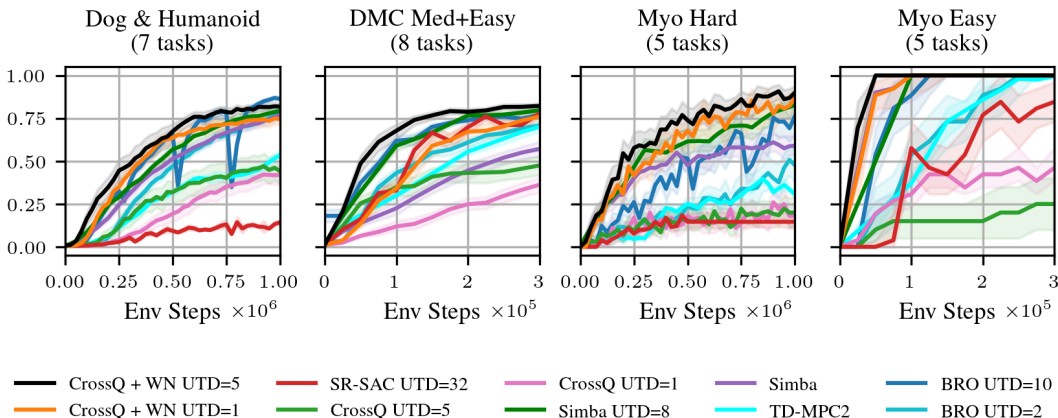

Figure 6: *CrossQ* WN UTD *scaling behavior.* We plot the IQM return and $90\%$ stratified bootstrapped confidence intervals for different UTD ratios. The results are aggregated over 15 DMC environments and 10 random seeds each according to Agarwal et al. [1]. The sample efficiency scales reliably with increasing UTD ratios.

# E    Per environment learning curves

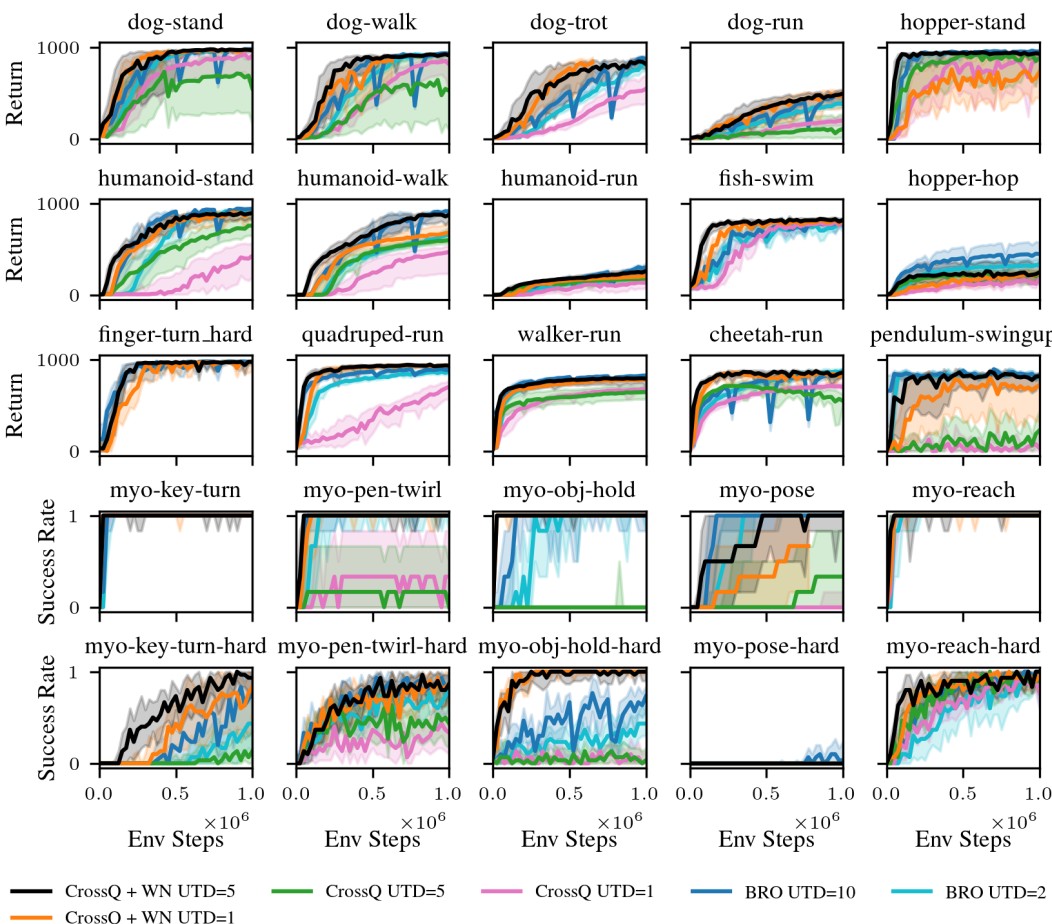

Figure 7: *CrossQ* WN + UTD=5 *against baselines.* We compare our proposed CrossQ + WN UTD=5 against two baselines, BRO [32] and SR-SAC UTD=32. Results are reported on all 15 DMC and 10 MyoSuite tasks. We plot the IQM and 90% stratified bootstrapped confidence intervals over 10 random random seeds. Our proposed approach proves competitive to BRO and outperforms the CrossQ baseline. We want to note that our approach achieves this performance without requiring any parameter resetting or additional exploration policies.

