# OpenReview forum: "Scaling Off-Policy Reinforcement Learning with Batch and Weight Normalization"
_NeurIPS.cc/2025/Conference — NeurIPS 2025 poster_

### Official Review · Reviewer_z4vh · 2025-06-08

**Clarity:** 3
**Significance:** 3
**Originality:** 2
**Rating:** 5
**Confidence:** 5

**Summary:**

The paper studies the updates-per-data (UTD) scaling properties of the recent CrossQ algorithm. The authors find that the vanilla CrossQ formulation does not scale reliably with UTD. They propose design changes that improve its scaling properties (i.e., using weight normalization and bringing back the target network). The authors test their approach on 25 simulated locomotion and manipulation tasks of varying complexity and find that incorporating these design choices improves not only the UTD scaling of CrossQ but also decreases the hyperparameter sensitivity of the algorithm.

**Questions:**

1. In Figure 5, what network size does SAC + LN + WN use? Do authors have any hypothesis as to why this version performs so poorly on MyoSuite given that Simba/BRO/CrossQ perform well (and all are driven by size+normalization)?
2. Do authors have any insights into why target networks are important in their proposed version of CrossQ? Does the importance of the target network correlate with higher UTD?
3. Do the authors have any opinion on the generality of the proposed architecture? For example, could the proposed architecture (BN + WN) be used in BRO instead of BRONet?

**Ethical Concerns:**

["NO or VERY MINOR ethics concerns only"]

**Final Justification:**

Despite the hesitation of other reviewers and limited novelty, I believe that this work will be interesting and helpful for the community. As such, I recommend acceptance.

**Limitations:**

yes

**Quality:**

3

**Strengths And Weaknesses:**

Strengths:

1. The paper is very clearly written and delivers on the claims made in the introduction.
2. Understanding compute scaling in off-policy RL is an active and promising research area
3. The proposed method is simple and seems to be effective

Weaknesses:

1. Slightly limited novelty of the proposed approach. But still, I think the result is useful


Nitpicks/Editorial:

1. (109) "Janner et al. [21] generate additional modeled data, by virtually increasing the UTD. In REDQ, Chen et al. [8] leverage ensembles of Q-functions, while Hiraoka et al. [17] use dropout and LN to effectively scale to higher UTD ratios" - the way MBPO is described sounds odd (it uses world model to simulate transitions that are used in high UTD training); description of RedQ also slightly misses the point (the sampling from the ensemble is important there)
2. (165) "which in turn decreases the ELR even quicker and could be the case for instabilities and low training performance" - "instabilities" paired with very low ELR sounds a bit odd.

---

> ### Author Rebuttal · Authors · 2025-07-31
>
> We thank the reviewer for their very positive feedback and their additional questions and comments to improve the paper further.
>
> We were especially happy to see that they found the **paper clearly written** and acknowledged that we **deliver on the claims made in the paper**. Further, we were happy to read that they acknowledge the simplicity and effectiveness of our proposed solution and that we **contribute useful results to a promising research area**.
>
> We thank the reviewer for pointing out the nitpicks, we will make sure to improve those points in the final version of the paper.
>
> Regarding the reviewer’s question:
>
> ---
>
> ### **“In Figure 5, what network size does SAC + LN + WN use?”**
>
> For the ablation in Figure 5, all methods use the exact same critic network size (MLP with two hidden layers of size 512 neurons each), in order to isolate the respective changes as cleanly as possible.
>
> ---
>
> ### **“Do authors have any hypothesis as to why this version performs so poorly on MyoSuite?”**
>
> As to why SAC+LN+WN performs worse is a very interesting question about which, we can only hypothesise for now.
>
> First, for SAC+LN+WN, we do see a performance improvement between UTD 1 and 5 in Figure 5 on Myo Hard from ~0.3 to ~0.6. For BRO we have two data points, UTD=2 and 10, if we assume a linear trend, then SAC+LN+WN would only be slightly worse than BRO on Myo Hard. Of course to confirm this we would need to run the BRO baseline for those UTDs, however, it shows that the performance gap for the SAC+LN+WN to the LN based baselines (BRO and Simba) might not be as large as it might look at first sight.
>
> Second, the remaining performance delta of SAC+LN+WN towards BRO/Simba can potentially be attributed to the much simpler and smaller architecture compared to the BRO/Simba baselines (~600k vs. ~5M parameters) which include skip connections and additional implementation details.
>
> Third, the SAC+LN+WN baseline demonstrates higher sample- and parameter-efficiency of BN compared to LN in this context. As such, our results reproduce the findings from the original CrossQ paper but in the high UTD and WN setting. So far those findings are empirical and we believe further understanding this on a theoretical level would be a really interesting endeavour for future work.
>
> ---
>
> ### **Do authors have any insights into why target networks are important in their proposed version of CrossQ?**
> In general the importance of the target network in off-policy RL is a really interesting question. In our specific case, Figure 5 shows that in general CrossQ+WN also manages to learn without a target network, however, the performance is significantly reduced. Considering the constant, but overall higher effective learning rate of CrossQ+WN compared to vanilla CrossQ (Figure 3), we think that the target network still adds a stabilizing effect to the training dynamics that ultimately increases the overall learning performance. At least in our experiments we did not find a meaningful correlation between UTD and the importance of the target network. However, it is a really interesting question that deserves a much deeper investigation in future work.
>
> ---
>
> ### **Generality of the proposed BN+WN solution**
> Of course, technically nothing speaks against replacing BroNet with a BN+WN architecture. We find that our proposed solution is competitive to BRO. Apart from the larger and more complicated network architecture, BRO is also based on SAC and has many additional implementation details like resetting and a second optimistic actor. Both are orthogonal to our proposed method. In our ablations we already show that our method neither requires, nor significantly benefits from additional network resets. As such, integration with BRO would investigate whether CrossQ+WN would additionally benefit from better exploration through BRO’s optimistic actor. Of course, this is an interesting question. Due to limited computational resources and time during the rebuttal, and since most reviewers asked about the generality of our approach w.r.t. "more basic" RL algorithms such as TD3, we chose to focus on extending our results to TD3.
>
> For the rebuttal we ran TD3+BN+WN experiments on the DMC dog and humanoid environments and Myo Hard for different UTDs (a total of 12 different environments). We get results that are comparable to our CrossQ+WN experiments  without any tuning of hyperparameters. As such, we can confirm that the **sample-efficiency and stable UTD scaling benefits of BN+WN extend to TD3 and beyond CrossQ**. As per the new NeurIPS rebuttal guidelines we were not allowed to share a figure during the rebuttal. However, we will add the additional TD3 results to the final paper.

---

> > ### Comment · Reviewer_z4vh · 2025-07-31
> > **.**
> >
> > Thank you for answering my questions. My final proposition is that authors implement slight changes in the manuscript to reflect our discussion and make these issues clearer for the reader.

---

> > > ### Author Response · Authors · 2025-08-01
> > >
> > > We thank the reviewer again, and we agree to make the requested slight changes to improve our manuscript.

---

> > > > ### Comment · Reviewer_z4vh · 2025-08-01
> > > > **Thank you for the rebuttal**
> > > >
> > > > Thank you for the rebuttal - my concerns are addressed.
> > > >
> > > > While the novelty of this work is limited, I strongly believe that the presented results will be helpful for the RL community - especially given the recent interest in how normalization/regularization affects the scaling and performance of value-based agents. As such, I recommend acceptance.

---

### Official Review · Reviewer_Gun3 · 2025-06-27

**Clarity:** 3
**Significance:** 3
**Originality:** 2
**Rating:** 5
**Confidence:** 3

**Summary:**

The paper presents a modification to CrossQ, a strong RL algorithm based on SAC that cleverly uses batch normalization to imrprove performance while allowing one to remove the target network, where by adding weight normalization one can increase the update-to-data ratio, increasing performance while keeping the previous sample efficiency.

Furthermore the authors show that it's possible to add back the target network to CrossQ + Weight Normalization, and that it gives them a further improvement in performance.

**Questions:**

Do you believe Weight Normalization could improve the scaling of UTD of other RL algorithms? Would it be possible to test a few of them?

**Ethical Concerns:**

["NO or VERY MINOR ethics concerns only"]

**Final Justification:**

The paper explores the impact of Weight Normalization on the performance of off-policy RL algorithms for continuous control, adding another result to our understanding of the interplay between Neural Network architecture and final performance in Reinforcement Learning tasks.

**Limitations:**

yes

**Quality:**

3

**Strengths And Weaknesses:**

The paper is clear about its contribution and motivation, the results are compelling and the experiments are done over a good variety of environments, the presented method is simple while having a significant effect.

The main weakness is that the paper focuses on augmenting a single specific RL algorithm/method, and doesn't really evaluate whether Weight Normalization could be beneficial for other algorithms like a more basic SAC, TD3, PPO or others.

---

> ### Author Rebuttal · Authors · 2025-07-31
>
> We thank the reviewer for their positive feedback and their additional questions.
>
> We were especially happy to see that the reviewer called our results compelling and acknowledged the simplicity and significant effect of our proposed method.
>
> Regarding the reviewer’s question:
>
> ---
>
> ### **“Do you believe Weight Normalization could improve the scaling of UTD of other RL algorithms? Would it be possible to test a few of them?”**
>
> For clarity, we want to answer this question in several steps:
>
> **Could WN alone be beneficial for scaling other more basic algorithms such as SAC, TD3, PPO?**
>
> Our paper focuses on off-policy RL, as such we believe that PPO is a bit out of scope. Of course it would be interesting to test, however, in on-policy deep RL the state-based critic acts as a baseline for the advantage estimation, and therefore plays a less critical role compared to off-policy RL where the state-action critic alone is used to update the policy. As such we will dismiss PPO here since on-policy scaling would deserve its own independent study.
>
> We see the core contribution of our paper in showing that the combination of BN+WN leads to stable scaling across different UTDs. We provide analysis as to the underlying reason (exploding weight norms for the CrossQ baseline vs. stable effective learning rate for CrossQ+WN). The justification to apply WN is based on the scale invariance that BN brings to the network. As such, we need to differentiate between “just WN” and the combination of BN+WN.
>
> **Testing BN+WN with other more basic algorithms**
>
> We want to clarify that all experiments in the paper are based on SAC as the base algorithm, since CrossQ+WN == SAC+BN+WN.
>
> For the rebuttal we ran TD3+BN+WN experiments on the DMC dog and humanoid environments and Myo Hard for different UTDs (a total of 12 different environments). We get results that are comparable to our CrossQ+WN experiments  without any tuning of hyperparameters. As such, we can confirm that the **sample-efficiency and stable UTD scaling benefits of BN+WN extend to TD3 and beyond CrossQ**. As per the new NeurIPS rebuttal guidelines we were not allowed to share a figure during the rebuttal. However, we will add the additional TD3 results to the final paper.

---

> > ### Author Response · Authors · 2025-08-01
> >
> > We would like to share a table with our new TD3 + BN + WN results.
> >
> > Below, you see IQM Return/Success and the upper and lower bounds of the 90% bootstrap confidence intervals, analogous to Figure 4 in the paper. Here we present results for TD3 + BN + WN and a TD3 + BN baseline on DMC Dog & Humanoid and the Myo Hard environments, which is a total of 12 environments. First, we see that TD3 + BN + WN significantly outperforms the TD3 + BN baseline everywhere by a large margin. Second, increasing the UTD from 1 to 5 stably scales the sample efficiency. As such, the TD3 + BN + WN experiment reproduces our CrossQ + WN findings of increasing sample efficiency and scaling stably with increasing UTD.
> >
> > ---
> >
> > **DMC Dog & Humanoid** @ 1M (IQM Return and 90% bootstrap CI)
> >
> > |           | UTD=1              | UTD=5              |
> > |-----------|--------------------|--------------------|
> > | **TD3+BN+WN** | **0.78** (0.75 - 0.79) | **0.81** (0.79 - 0.82) |
> > | TD3+BN    | 0.67 (0.63 - 0.71) | 0.68 (0.63 - 0.72) |
> >
> > ---
> >
> > **Myo Hard** @ 1M (IQM Success and 90% bootstrap CI)
> >
> > |           | UTD=1              | UTD=5              |
> > |-----------|--------------------|--------------------|
> > | **TD3+BN+WN** | **0.65** (0.58 - 0.71) | **0.78** (0.68 - 0.85) |
> > | TD3+BN    | 0.48 (0.40 - 0.57)  | 0.65 (0.58 - 0.72) |

---

> > > ### Comment · Reviewer_Gun3 · 2025-08-04
> > >
> > > Thank you for the aditional results, I believe there's still a lot to understand on the effect of architectural choices on RL performance, but this paper does bring some new interesting datapoints. I'll bump my rating to accepted, but would ask the authors to add a comment about "On the consistency of hyper-parameter selection in value-based deep reinforcement learning" to the Related Work section, as that paper has previously analyzed the impact of different normalization on the performance of RL algorithms.

---

> > > > ### Author Response · Authors · 2025-08-04
> > > >
> > > > Thank you for increasing your score! We will include the requested reference and comment in our related work section.

---

### Official Review · Reviewer_uACo · 2025-06-30

**Clarity:** 3
**Significance:** 2
**Originality:** 2
**Rating:** 4
**Confidence:** 4

**Summary:**

This work extends CrossQ by integrating weight normalization (WN) to stabilize training at high update-to-data ratios. As a result, it achieves superior sample efficiency on continuous-control benchmarks and delivers competitive performance without requiring parameter resets.

**Questions:**

-  BN and WN are generic normalization techniques whose benefits may extend beyond CrossQ. Have you evaluated their effectiveness across other network architectures, design choices, or reinforcement learning algorithms to demonstrate their broader applicability?
- I appreciate the comprehensive ablation study. Would you also consider evaluating the effect of applying WN at various layers within CrossQ, rather than only in the first two linear layers, to clarify how WN placement influences performance?

**Ethical Concerns:**

["NO or VERY MINOR ethics concerns only"]

**Final Justification:**

Since the authors have addressed all my questions and concerns, I have raised my score accordingly.

**Limitations:**

The authors have already noted several limitations. Additional limitations are discussed in the Questions section.

**Paper Formatting Concerns:**

No major formatting issues were found.

**Quality:**

3

**Strengths And Weaknesses:**

### Strengths:

- Offers clear theoretical explanations for why WN is effective.

-  The presentation is clear and easy to follow.

-  Includes extensive hyperparameter ablation studies.

### Weaknesses:

-  BN and WN are general-purpose normalization techniques that could be applied to other designs/algorithms, yet this study evaluates them only on CrossQ[1].

[1]CrossQ: Batch normalization in deep reinforcement learning for greater sample efficiency and simplicity, Bhatt et al, ICLR 2024.

-  Figure 4 indicates that increasing the UTD ratio from 1 to 20 for CrossQ + WN yields no significant gains on the DMC medium + easy and Myo hard tasks, suggesting that  CrossQ + WN merely preserves rather than improves the performance when scaling up the UTD ratio, while increasing the compute cost by roughly 20 times.

---

> ### Author Rebuttal · Authors · 2025-07-31
>
> We want to thank the reviewer for the positive feedback and their valuable questions.
>
> We were especially happy that they recognized the **superior sample efficiency** of our approach and positively mentioned the **clear theoretical explanations why WN is effective**, **clear and easy to follow** presentation and our **extensive hyperparameter ablations**.
>
> Regarding their questions, we believe we have been able to address their points here.
>
> ---
>
> ### **Does the proposed BN+WN solution show benefits beyond CrossQ (network architectures, design choices, or reinforcement learning algorithms)**
>
> For the rebuttal, we ran TD3+BN+WN experiments on the DMC dog and humanoid environments and Myo Hard for different UTDs (a total of 12 different environments). We get results that are comparable to our CrossQ+WN experiments without any tuning of hyperparameters. As such, we can confirm that the **sample-efficiency and stable UTD scaling benefits of BN+WN extend to TD3 and beyond CrossQ**. As per the new NeurIPS rebuttal guidelines, we were not allowed to share a figure during the rebuttal. However, we will add the additional TD3 results to the final paper.
>
> Further, our proposed BN+WN approach is agnostic to common design decisions (e.g., network width and depth). In the paper’s ablations, we test algorithmic design decisions like resetting and the removal of target networks. For some other design decisions like skip connections or different activation functions, one would need to ensure to preserve scale invariance w.r.t. parameter rescaling when applying those. Here we also want to mention that we did not specifically tune the architecture beyond our proposed BN+WN. For the architecture, we use a standard MLP with 2 hidden layers, which results in a very simple algorithm that has been positively pointed out by most reviewers. That being said, we will extend the final paper with some architectural ablations.
>
> ---
>
> ### **“Would you also consider evaluating the effect of applying WN at various layers within CrossQ, rather than only in the first two linear layers, to clarify how WN placement influences performance?”**
>
> We believe there is a slight misunderstanding, for which we need to improve the formulation in the paper. We write that “We apply WN to the first two linear layers [in the network]”. Since we only use MLPs with two hidden layers, this means that we apply WN to all layers but the final linear output layer.
>
> The reason for not placing WN on the output layer is to allow scaling of predicted Q values to arbitrary magnitudes and crucially avoid the risk of underfitting. Additionally, layers that are not followed by BN are not scale invariant w.r.t. re-scaling their weights.
>
> The reason for applying WN to all hidden layers is to preserve a constant effective learning rate within the network, since the hidden layers are followed by BN. We hope this clarifies the open question and we will make sure to improve the formulation in the paper to remove the ambiguity.
>
> That being said, we are more than happy to prepare additional ablation studies on the placement of WN within the network to the final paper.
>
> ---
>
> ### **UTD scaling on “DMC medium + easy and Myo hard tasks”**
> We assume the reviewer meant to refer to Myo Easy and Myo Hard instead of DMC Med+Easy, if not we apologize.
>
> We believe that the important contribution of BN+WN is that it **stabilizes training across scaling UTDs**. As such, UTD is not a brittle hyperparameter anymore, but performance stays at least constant or increases. While increasing the UTD hyperparameter may improve sample efficiency, there is no theoretical guarantee that increasing the UTD will always improve the downstream performance of the policy, as such stability is an important aspect. For instance on Myo Easy, the vanilla CrossQ baselines completely deteriorate in performance when increasing the UTD, which our method mitigates.
>
> On Myo Easy, CrossQ+WN already reached the maximum score at UTD=1. So, on those tasks higher UTDs confirm the stability across UTDs in contrast to vanilla CrossQ.
> On Myo Hard, our method also remains stable across all UTDs and outperforms every baselines method at every UTD.
> Finally, on DMC Med+Easy and the Dog & Humanoid environments CrossQ+WN also scales stably and its sample efficiency significantly benefits from increasing the UTD.

---

> > ### Comment · Reviewer_uACo · 2025-07-31
> >
> > Thank you for your response. Although the NeurIPS rebuttal guidelines prevent sharing the new results (TD3+BN+WN) with figures, could you please present the results in a table?

---

> > > ### Author Response · Authors · 2025-08-01
> > >
> > > Yes, of course, we are happy to share the new TD3 + BN + WN results. In the tables below, you see IQM Return/Success and the upper and lower bounds of the 90% bootstrap confidence intervals, analogous to Figure 4 in the paper. Here we present results for TD3 + BN + WN and a TD3 + BN baseline on DMC Dog & Humanoid and the Myo Hard environments, which is a total of 12 environments. First, we see that TD3 + BN + WN significantly outperforms the TD3 + BN baseline everywhere by a large margin. Second, increasing the UTD from 1 to 5 stably scales the sample efficiency. As such, the TD3 + BN + WN experiment reproduces our CrossQ + WN findings of increasing sample efficiency and scaling stably with increasing UTD.
> > >
> > > ---
> > >
> > > **DMC Dog & Humanoid** @ 1M (IQM Return and 90% bootstrap CI)
> > >
> > > |           | UTD=1              | UTD=5              |
> > > |-----------|--------------------|--------------------|
> > > | **TD3+BN+WN** | **0.78** (0.75 - 0.79) | **0.81** (0.79 - 0.82) |
> > > | TD3+BN    | 0.67 (0.63 - 0.71) | 0.68 (0.63 - 0.72) |
> > >
> > > ---
> > >
> > > **Myo Hard** @ 1M (IQM Success and 90% bootstrap CI)
> > >
> > > |           | UTD=1              | UTD=5              |
> > > |-----------|--------------------|--------------------|
> > > | **TD3+BN+WN** | **0.65** (0.58 - 0.71) | **0.78** (0.68 - 0.85) |
> > > | TD3+BN    | 0.48 (0.40 - 0.57)  | 0.65 (0.58 - 0.72) |

---

> > > > ### Comment · Reviewer_uACo · 2025-08-01
> > > >
> > > > Thank you for providing the results. I have no further questions, and I have increased my score accordingly.

---

### Official Review · Reviewer_htkW · 2025-06-30

**Clarity:** 3
**Significance:** 2
**Originality:** 2
**Rating:** 5
**Confidence:** 3

**Summary:**

* In this paper, the authors point out that the existing CrossQ framework shows high sample efficiency at low update-to-data ratio (UTD=1), but becomes unstable in a high UTD environment. To this end, by incorporating a weight normalization technique on the CrossQ framework (i.e., CrossQ + WN), the authors aim to suppress the explosive increase of parameter norm during training and improve the learning stability even at high UTD. By introducing a simple but effective method, the authors prove that it is more effective than existing works in various continuous state-action benchmarking tasks.

**Questions:**

* The authors applied weight normalization only to the first two linear layers. I wonder if there are any ablation studies or experimental results on where to apply WN.
* In Fig. 5, what baseline should we interpret no target network, no WN based on? And is there any experimental result when LN/WN are excluded from SAC?

**Ethical Concerns:**

["NO or VERY MINOR ethics concerns only"]

**Final Justification:**

Based on my initial review and the author's feedback, I maintain a positive assessment of this paper.

**Limitations:**

* I think it is a meaningful study in terms of practicality. However, the novelty would be ambiguous in terms of technical algorithms or theoretical advances of reinforcement learning.
* It can be considered as close to fine-tuning through the introduction of WN in the existing algorithm CrossQ, so the technical novelty may be limited.
* I wonder whether the proposed method can achieve the same effect when applied to other reinforcement learning frameworks (e.g., SAC, PPO, DDPG) rather than CrossQ.

**Paper Formatting Concerns:**

There are no formatting concerns.

**Quality:**

3

**Strengths And Weaknesses:**

**[Strengths]**
* One of the advantages of the proposed CrossQ+WN is that training is stable even in high UTD environments and that it achieves high performance with minimal computational resources.
* The authors propose a relatively simple but effective algorithm without additional exploration policies, parameter resets, etc.
* In particular, the number of parameters of the proposed method is 600k, which is very parameter-efficient compared to the existing BRO with 5M parameters.
* No per-task tuning is required.

**[Weaknesses]**
* The performance improvement is based on empirical observations, and it would be more valuable if mathematical analysis related to convergence or sample efficiency were included.
* As the authors mentioned, the quality of the paper would be improved in terms of generalizability if the proposed method is applied and its performance is verified in a discrete state-action space or vision-based tasks.

---

> ### Author Rebuttal · Authors · 2025-07-31
>
> We thank the reviewer for their positive feedback and their additional questions.
>
> We were especially happy to read that the reviewer **acknowledges the simplicity and effectiveness** of our proposed approach, as well as its **parameter-efficiency** and that they mention it is a **meaningful study in terms of practicality**.
>
> Regarding the reviewer’s open questions, we address their points here.
>
> ---
>
> ### **Does the proposed method show the same effect when applied to other RL algorithms beyond CrossQ?**
>
> Regarding the requested SAC experiments, we want to clarify that all experiments in the paper are based on SAC as the base algorithm, since CrossQ+WN == SAC+BN+WN.
>
> For the rebuttal, we ran TD3+BN+WN experiments on the DMC dog and humanoid environments and Myo Hard for different UTDs (a total of 12 different environments). We get results that are comparable to our CrossQ+WN experiments without any tuning of hyperparameters. As such, we can confirm that the **sample-efficiency and stable UTD scaling benefits of BN+WN extend to TD3 and beyond CrossQ**. As per the new NeurIPS rebuttal guidelines, we were not allowed to share a figure during the rebuttal. However, we will add the additional TD3 results to the final paper.
>
> With respect to PPO, our paper focuses on off-policy RL, as such we believe that PPO is a bit out of scope. Of course it would be interesting to test, however, in on-policy deep RL the state-based critic acts as a baseline for the advantage estimation, and therefore plays a less critical role compared to off-policy RL where the state-action critic alone is used to update the policy. As such we hypothesise that in general it won't benefit as much from the proposed changes to the critic.
>
> ---
>
> ### **“The authors applied weight normalization only to the first two linear layers. I wonder if there are any ablation studies or experimental results on where to apply WN?”**
>
> We believe there is a slight misunderstanding, for which we need to improve the formulation in the paper. We write that “We apply WN to the first two linear layers [in the network]”. Since we only use MLPs with two hidden layers, this means that we apply WN to all layers but the final linear output layer.
>
> The reason for not placing WN on the output layer is to allow scaling of predicted Q values to arbitrary magnitudes and crucially avoid the risk of underfitting. Additionally, layers that are not followed by BN are not scale invariant w.r.t. re-scaling their weights.
>
> The reason for applying WN to all hidden layers is to preserve a constant effective learning rate within the network, since the hidden layers are followed by BN. We hope this clarifies the open question and we will make sure to improve the formulation in the paper to remove the ambiguity.
>
> That being said, we are more than happy to prepare additional ablation studies on the placement of WN within the network to the final paper.
>
> ---
>
> ### **“In Fig. 5, what baseline should we interpret no target network, no WN based on?”**
> We apologize for the unclear caption and we will make sure to improve it as well as the main text to make it more clear. Both are ablations based on our proposed CrossQ + WN algorithm. As such, these two baselines refer to “CrossQ + WN without a target network” and “CrossQ + WN without WN” respectively. The latter would be the same as saying “CrossQ with a target network”. As such, we ablate the role of the target network and the role of WN in our proposed method.
>
> ---
>
> ### **“is there any experimental result when LN/WN are excluded from SAC”**
> We assume that this question is referring to the SAC+LN+WN baseline? Since CrossQ+WN == SAC+BN+WN, our goal with that particular baseline was to ablate the effect of BN in our proposed algorithm over another normalization method that leads to scale invariance. We find BN to be advantageous. Should we have misunderstood the question, please let us know.

---

> > ### Author Response · Authors · 2025-08-01
> >
> > We would like to share a table with our new TD3 + BN + WN results.
> >
> > Below, you see IQM Return/Success and the upper and lower bounds of the 90% bootstrap confidence intervals, analogous to Figure 4 in the paper. Here we present results for TD3 + BN + WN and a TD3 + BN baseline on DMC Dog & Humanoid and the Myo Hard environments, which is a total of 12 environments. First, we see that TD3 + BN + WN significantly outperforms the TD3 + BN baseline everywhere by a large margin. Second, increasing the UTD from 1 to 5 stably scales the sample efficiency. As such, the TD3 + BN + WN experiment reproduces our CrossQ + WN findings of increasing sample efficiency and scaling stably with increasing UTD.
> >
> > ---
> >
> > **DMC Dog & Humanoid** @ 1M (IQM Return and 90% bootstrap CI)
> >
> > |           | UTD=1              | UTD=5              |
> > |-----------|--------------------|--------------------|
> > | **TD3+BN+WN** | **0.78** (0.75 - 0.79) | **0.81** (0.79 - 0.82) |
> > | TD3+BN    | 0.67 (0.63 - 0.71) | 0.68 (0.63 - 0.72) |
> >
> > ---
> >
> > **Myo Hard** @ 1M (IQM Success and 90% bootstrap CI)
> >
> > |           | UTD=1              | UTD=5              |
> > |-----------|--------------------|--------------------|
> > | **TD3+BN+WN** | **0.65** (0.58 - 0.71) | **0.78** (0.68 - 0.85) |
> > | TD3+BN    | 0.48 (0.40 - 0.57)  | 0.65 (0.58 - 0.72) |

---

> > > ### Comment · Area_Chair_eFQC · 2025-08-04
> > >
> > > Dear reviewer,
> > >
> > > Thank you for your efforts so far in reviewing this paper. The authors have answered your questions. Would you please check their reply and let them and the other reviewers know what you think about their response?
> > >
> > > Thank you,
> > > Area Chair

---

> > ### Comment · Reviewer_htkW · 2025-08-04
> >
> > Thank you for your clear answer to my question.
> > Regarding Q2, my intention was simply to compare the SAC+LN+WN results in Fig. 5 with the SAC results (excluding LN+WN) for comparison purposes. (This may not be significant for the overall performance comparison.)
> >
> > I still believe the proposed method is simple but effective. For the sake of clarity, I recommend reflecting the answers provided above in the final manuscript.

---

> > > ### Author Response · Authors · 2025-08-04
> > >
> > > We thank the reviewer for their response. We are glad we were able to answer their questions.
> > >
> > > As recommended, we will make sure to integrate the clarifications we made during the rebuttal into the final paper.

---

### Note · Authors · 2025-08-12

We want to thank all reviewers again for taking the time to review our paper, for the positive and valuable discussions and for acknowledging our paper’s contributions.

We are very happy that we were able to answer all of the reviewers’ questions to their satisfaction and that all reviewers voted positively towards accepting the paper. For the final version we are integrating the additional experiments, discussions and related work, as discussed during the rebuttal.

To summarize, we will
- (i) add the additional rebuttal results confirming that the benefits of our proposed BN+WN also extend to other algorithms like TD3;
- (ii) clarify the formulations in the paper w.r.t. the placement of WN and the caption of Fig 5.;
- (iii) include the suggested additional ablations; and
- (iv) incorporate the suggested related work, minor edits and reflect the discussions during the rebuttal.


The overall takeaway is now clearer and stronger: combining BN with WN offers a simple, practical route to increasing sample efficiency of off-policy RL methods via stable high-UTD scaling without requiring larger models or resets.

Thank you again to everyone! We appreciate the constructive dialogue and believe the final version reflects it.

---

### Decision · Program_Chairs · 2025-09-17

**Decision:**

Accept (poster)

**Comment:**

The paper proposes adding Weight Normalization (WM) to the CrossQ RL algorithm, showing that it stabilizes the training and improves its performance, especially in the high update-to-data (UTD) ratios.

All reviewers are supportive of this work. We have four Accepts (5) and one Borderline Accept (4). Therefore, I also recommend acceptance of the paper.

Additional comments:
- Figures 1 and 2 appear without any description until much later in the paper. The first time a figure is explained is on page 4, and that is for Figure 3.

- Bellmann --> Bellman

- L67: It is not accurate to say that SAC solves the optimization problem indicated in the first few lines of Section 2. The optimization problem solved by SAC has an entropic term. And as a result, the state-value function (Eq. 1) is also defined differently in SAC. Please clarify.